# Plant Parasitic Nematodes: A Review on Their Behaviour, Host Interaction, Management Approaches and Their Occurrence in Two Sites in the Republic of Ireland

**DOI:** 10.3390/plants10112352

**Published:** 2021-10-30

**Authors:** Anusha Pulavarty, Aoife Egan, Anna Karpinska, Karina Horgan, Thomais Kakouli-Duarte

**Affiliations:** 1Molecular Ecology and Nematode Research Group, enviroCORE, Department of Science and Health, Institute of Technology Carlow, Kilkenny Road, R93 V960 Carlow, Ireland; anusha.pulavarty@itcarlow.ie (A.P.); aoife.eg@gmail.com (A.E.); anna.karpinska@itcarlow.ie (A.K.); 2Alltech Bioscience Centre, A86 X006 Dunboyne, County Meath, Ireland; khorgan@alltech.com

**Keywords:** plant parasitic nematodes, microbial, fermentation, bionematicides

## Abstract

Plant parasitic nematodes are a major problem for growers worldwide, causing severe crop losses. Several conventional strategies, such as chemical nematicides and biofumigation, have been employed in the past to manage their infection in plants and spread in soils. However, the search for the most sustainable and environmentally safe practices is still ongoing. This review summarises information on plant parasitic nematodes, their distribution, and their interaction with their host plants, along with various approaches to manage their infestations. It also focuses on the application of microbial and fermentation-based bionematicides that have not only been successful in controlling nematode infection but have also led to plant growth promotion and proven to be environmentally safe. Studies with new information on the relative abundance of plant parasitic nematodes in two agricultural sites in the Republic of Ireland are also reported. This review, with the information it provides, will help to generate an up-to-date knowledge base on plant parasitic nematodes and their management practices.

## 1. Introduction

There are nearly 4100 species of plant parasitic nematodes (PPN) reported to date that are considered to be a serious constraint for global food security [1]. Considering economic development and food preferences, the World Bank in 2008 estimated a 35% increase in world population by 2050, which will correspond to a 75% increase in food demand [2]. Therefore, it has become an environmental concern for relevant stakeholders worldwide to promote sustainable methods to enhance the efficiency of resource use [3].

The objectives of this review are to (1) report on the global PPN distribution along with studies performed in two Irish agricultural sites, (2) summarise information on host-parasite interactions and (3) discuss current PPN management strategies with an emphasis on the benefits of microbial and fermentation based products.

## 2. Plant Parasitic Nematode Species and Their Distribution

In the UK alone, it is estimated that the cyst nematodes *Globodera rostochiensis* and *Globodera pallida* are responsible for approximately 9% of the total UK potato production losses [4]. *Globodera pallida* is the predominant potato cyst nematode (PCN) that is found in more than 90% of the nematode-infested fields of England and Wales, with an overall estimated cost of £26 million per year in crop yield [5] and an additional £10 million per year in nematicides [6]. In tropical and sub-tropical climates, 14.6% crop losses primarily occur due to nematodes, whereas these are estimated to be 8.8% in developed nations [1]. Fleming et al. [7] reported the prevalence and diversity of PPN in the cereals and grasslands of Northern Ireland; *Meloidogyne* spp., *Heterodera* spp. and *Pratylenchus* spp. were found above the threshold levels for economic damage. This increase in nematode populations was reported due to poor cropping practices and climate change [7]. Surveys conducted in 35 states of the USA reported a total of 25% crop loss due to PPN [8]. Loss in crop productivity due to soybean cyst nematodes have been calculated to be around $US1.5 billion each year in the USA alone [9]. Under favourable environmental conditions, cereal cyst nematodes (CCN) can destroy 90% of crop fields [1]. Potato cyst nematodes cause a 9% loss of total potato production worldwide [1]. These nematodes, originating from South America, are currently major quarantine pathogens and have widely spread to all the potato-growing regions of the world [10].

Nose and Shiraishi [11] have reported a total loss of 2 × 10^6^ m^3^ of timber in Japan due to *Bursaphelenchus xylophilus* which has the potential to infect and kill all the pine trees in an infested area under favourable conditions. The Department of Agriculture, Food and the Marine (DAFM) in Ireland reported *B. xylophilus*, *G. pallida*, *G. rostochiensis*, *Meloidogyne chitwoodi* and *Meloidogyne fallax* as European Union quarantine pests [12].

## 3. Irish Case Studies

Previously there were reports on the occurrence of predatory nematodes [13] and insect parasitic nematodes [14] across Ireland, but to the best of our knowledge there are no reports on the occurrence/distribution of PPN in the Republic of Ireland. We have made an effort to report on the abundance of PPN in two agricultural sites in Ireland (Figure 1a). The findings in this review could predict the PPN species that could emerge as a potential threat to Irish crops in the future. Hence, this information would help relevant stakeholders and policy-makers to formulate policies on preventing the spread of these devastating organisms in the future.

Case study 1. Occurrence of PPN at Teagasc, Johnstown

The presence of PPN was recorded through field trials performed at Teagasc, Johnstown Castle, Wexford, Ireland (Figure 1b). This study was performed as a part of the ReNu2Farm project (www.nweurope.eu/renu2farm, accessed on 6 January 2021), which aims to increase the application of recycling-derived fertilisers (RDFs) on agricultural fields in North-West Europe (NWE) (unpublished). The novel fertilisers were environmentally risk-assessed using nematodes as bioindicators, and this study recorded the PPN reported here. The sampling took place in July 2019 while the temperature was in the range of 17–24 °C.

Total terrestrial nematode DNA was extracted from the soil subsamples using the Qiagen DNeasy PowerSoil Pro kit, as per the manufacturer’s instructions. The extracted DNA from soil samples was sent for sequencing (Novogene Ltd., Singapore) for amplification and identification. The data was obtained through amplicon sequencing on an Illumina paired-end platform of the nematode 18S V4 region using suitable primers from Bhadury et al. [15]. The cleaned sequences were clustered into molecular operational taxonomic units (MOTUs) based on a sequence similarity threshold of 97%. MOTUs are sequence clusters used as units of taxonomic diversity. The SILVA (release 138 SSURef_NR99) database was used for further taxonomic assignation of clustered MOTUs.

The sequences of the 18S rRNA gene have been determined for various nematode taxa including PPN, which were found in the soil during the ecological risk assessment. This gene is universal and comparable in all nematode species and is frequently used to deduce deep phylogenetic relationships [16,17,18]. It has a relatively constant length, meaning genes from disparate taxa can be aligned quite easily. The SSU gene is present in 50–100 copies per cell and is thus easier to isolate or amplify by PCR than single copy genes. It contains a series of conserved and less-conserved regions. The less conserved regions can be used to identify nematodes to genus, family or species, while universal primers can be bioinformatically designed to this end in the conserved areas. There is a wealth of SSU genetic information held in online databases, as the gene has been used in attempts to establish a phylogeny for the Nematoda [17], and it can be used to aid the classification of nematodes with uncertain or suspect phylogeny [19].

The obtained data expressed as MOTUs, was used to calculate the relative abundance of identified nematode species present in the samples. The average relative abundance representing PPN taxa is summarised in Table 1. Relative abundance is the relative percentage quantity of the study organism in relation to the total number of organisms in a sample. The overall number of identified nematode species were 45, out of which 16 belong to PPN.

Case study 2. Occurrence of PPN taxa at Teagasc, Oakpark

Another case study was performed with soil samples collected from the rhizosphere of an oilseed rape (OSR, *Brassica napus*) crop, in Teagasc Oakpark, County Carlow, Republic of Ireland (Figure 2). This study was performed while analysing the suitability of nematode communities as bioindicators of OSR nickel bioremediation, by utilising a morphological assessment [20]. The sampling took place in August 2014 with an average temperature range of 10–18 °C. The soil was sampled from the environment, homogenised and divided into four parts as replications. Each replication consisted of 400 cm^3^ of soil. The nematodes were extracted from the soil using the decanting and sieving method. The nematodes were then suspended in Ludox, a silica colloidal solution and collected on a 40 µm sieve [21]. The extracted nematodes were preserved in DESS (dimethyl sulphoxide, disodium EDTA) and saturated with NaCl [22]. The preserved nematodes were mounted onto slides, and 50 nematodes per replication were identified under a high-power light microscope using the keys of Powers et al. [23]. The feeding types were assigned according to Yeates et al. [24], and by using NINJA, the nematode monitoring programme [25]. The mean relative abundance, a measure of species abundance (the number of individuals) relative to the abundance of other species, which is measured on a log scale, was also assigned (Table 2).

## 4. Nematode Behaviour, Feeding and Host–Parasite Interactions

Plant parasitic nematodes demonstrate a wide variety of interactions with their host. They can be categorized into ecto or endoparasites depending on the plant tissues they feed on. Some PPN are migratory as they easily move from soil to plant tissues, whereas others are sedentary with an adult female being completely immobile and stuck to the roots of the plant. The sedentary endoparasites feed with the help of specialised cells present around the female head. Most species of PPN have a needle-like protrusible oral structure called a stylet, which helps to puncture the host plant tissues. They release specific enzymes into the tissues that help in partially digesting the plant cells for easy ingestion into the nematode gut [26]. The migratory adult female usually deposits its eggs in soil or plants, based on its position. On the other hand, the most economically important PPN, such as root-knot (*Meloidogyne* spp.), cyst (*Globodera* spp., *Heterodera* spp.), reniform (*Rotylenchulus* spp.), and citrus (*Tylenchulus semipenetrans*) species are biotrophic, sedentary in nature [27]. They lay a large cluster of eggs either inside their bodies or attached as masses to their body. Within the egg after embryogenesis, first stage juveniles (J1) moult to form the second stage infective juveniles (J2) that hatch from the eggs to infect the root tissues. The expanded root parts appear as galls, containing root-knot nematode females, or as pathological nodes on roots, induced by traumatic wounds done by the nematode stylets or spears puncturing the root surface.

The genus *Meloidogyne* consists of nearly 98 species with a wide host range and can parasitize many vascular plants [28]. Second stage juveniles adopt both physical and enzymatic approaches to penetrate the host. With the help of a stylet, they damage the plant cell wall and then release cellulolytic and pectolytic enzymes to completely digest it. Conversely, the cyst nematodes display an upward intercellular movement within the root cells and try to reach the zone of differentiation, via the root tip, the apical meristems and the vascular cylinder. The giant cells, generated due to repeated nuclear divisions, in this region act as permanent feeding sites for the sedentary J2, where they undergo third and fourth stage moults to form reproductive male and female adults [29].

De Waele, D. and Elsen [30] have reported the difficulty in mitigating the damage caused by *Meloidogyne* species due to their short life cycle and broad host range. Due to short life cycles, these organisms spread faster and infect nearby crop plants within a short time. These nematode species are well adapted to flood conditions, with the potential to attack both upland and lowland rice [31] and can cause up to 85% crop loss [32]. *Meloidogyne incognita* have a unique set of putative genes that reduce plant immunity, detoxification and defence mechanisms and that help them to survive inside the plant host [33]. The plant parasitic lifestyle of root-knot nematodes is mainly due to the abundance of plant-cell-wall-degrading genes in their genomes [34]. *Meloidogyne* species, being obligate biotrophs, have to continuously suppress host defence mechanisms for the survival of their feeding structures. *Meloidogyne incognita* have a defence mechanism similar to that of plant pathogenic bacteria, in which they secrete calreticulin, which helps in sequestering free calcium ions and therefore curbs calcium ion influx [35]. Root knot nematodes also interact with other pathogens like *Fusarium* wilt, *Rhizoctonia solani* and *Thielaviopsis basicola,* leading to complex plant diseases [36]. *Meloidogyne* species have a gene that mimics the rhizobial *NodL* gene, which induces nodular formation in legumes [37]. Though the nematode exudates potentially alter nodulation signalling in legumes, some mutant legumes that do not support normal nodulation are tolerant to nematode infections [38].

Cyst nematodes are obligate biotrophs, and the most devastating species among them are: soybean cyst nematodes (SCN; *H. glycines*), PCN (*G. pallida* and *G. rostochiensis*) and CCN (including *Heterodera avenae* and *Heterodera filipjevi*). It is nearly impossible to eradicate potato cyst nematodes due to their prolonged survival of up to 20 years in the soil, even in the absence of a host, and their tolerance to extremely low temperatures [39]. The dormant second-stage juveniles (J2) hatch from the eggs in the presence of a host-derived chemical that is abundant in root diffusates [40]. The released second-stage juveniles invade the host intercellularly and reach the inner cortex. This juvenile has a peculiar behaviour; it keeps inserting its stylet into various cells until it finds a cell that does not collapse its protoplast and does not cover the stylet with a layer of callose- like material. Finally, it finds a suitable cell that becomes the initial syncytial cell (ISC). Subsequently, the cell walls of the ISC surrounding cells are dissolved and protoplasts fuse to form the large multinucleate feeding cell called the syncytium [29]. All the cells surrounding the ISC cells contribute to the formation of the syncytium, where DNA is synthesised and metabolism is enhanced to provide a nutrient-rich medium to the infecting nematode [41]. The nematode remains attached to the feeding site for several weeks, wherein it undergoes two further moults to form a complete adult [42]. The male adults remain vermiform and leave the root cells, whereas the female adults grow, get fertilised and finally die to form a tanned body wall that converts into a cyst, which bears the next generation of eggs [43]. Besides proteins that modify the host cell wall, many nematodes are reported to have effector molecules that suppress the host defensive mechanisms [44] and modify the host nucleus [43]. Significant efforts have been made to understand syncytium formation in the biology of cyst nematodes. These nematodes initiate the production of a peptide that has complete similarity with the plant peptide CLAVATA3 (CLE3). The stem cells in shoot and floral meristems of *Arabidopsis* secrete CLV3, which is the founding member of the CLE protein family, which eventually restricts the size of the stem cell population [45]. Therefore, indirect nematode manipulation of the CLAVATA signalling pathway induces the feeding site [45]. In addition, an effector that modulates the auxin flow pattern into the feeding structures has been identified [46]. Moreover, various genes expressed within the syncytia have been studied through microarray analysis to understand the basis of the feeding site formation [47].

Many taxa of nematodes produce specific secretions that contain effectors and cell-wall degrading enzymes, such as cellulases [48], pectate lyases [49] and xylanases [48]; these degrade the cell walls of infected cells. The effectors also have the potential to suppress the host immune system by altering its defence mechanisms. The species *Ditylenchus dipsaci* cause malformations in the infected plant tissues by withdrawing the cell contents through the nematode stylet [50]. This nematode has the unique property of resistance to dry conditions and freeze tolerance, due to the outer lipid layer of the fourth generation juvenile, which prevents water loss from its body [51]. The reniform nematodes have a unique way of interacting with their host plant. Initially, the adult female inserts one-third of its anterior body into the host root and establishes a feeding site to form a syncytium. After continuous feeding for about 2–3 days, the posterior part of the body outside the roots starts swelling near the vulval region to attain a kidney shape. Subsequently, within 7–9 days of thriving in environmentally favourable conditions, 40–100 eggs are laid within the gelatinous matrix produced by the uterine glands [52]. Under favourable conditions, PPN also interacts with other soil-borne pathogens like bacteria, fungi and viruses to suppress plants defence mechanisms or to cause a breakdown of plant resistance against infection [43].

## 5. Plant Responses to Nematode Infection

Based on plant cultivar and species, different plants react differently to nematode infections. Temperature, soil moisture content, nematode type, soil characteristics and crop rotations also affect the damage levels. Typical and most peculiar plant symptoms range from premature wilting, chlorosis, nutrient deficiency leading to stunted growth, fragile roots and swollen root areas due to gall formation. The *Pratylenchus* species cause lesions in roots leading to cell necrosis, browning and death, and root rotting due to secondary attachment by fungi and bacteria that thrive in soil. Infected plant roots undergo discolouration, and become stubby and stunted, making the plant susceptible to water stress conditions [53]. Some *Radopholus* spp. manifest as toppling disease in infected banana host plants [54]. Reduction in crop yield is often monitored as a sign of nematode infestation, both in terms of quality and quantity [55]. The threshold level for nematode infestation could be one nematode egg per 100 cm^3^ of soil [56]. The rice-stem nematodes, *Ditylenchus angustus*, feed ecoparasitically on the leaves and stems of rice and cause ufra disease in rice plants [54]. *Ditylenchus dipsaci* primarily infects onion and garlic, leading to the discoloration of the infected bulbs and the stunted growth of the host plants [50]. *Ditylenchus dipsaci* is a migratory endoparasite, whereas *Ditylenchus angustus* is a migratory ectoparasite. The host plant resistance (HPR) mode can be easily incorporated in the case of endoparasites, as they spend more than half of their life-cycle within the host. However, ectoparasitic nematodes cannot be strongly selected to develop HPR, due to their reduced specific feeding requirements. Different types of PPN and their mode of action are listed in Table 3.

## 6. Current Management Strategies and Their Limitations

Simply identifying the nematodes and applying the appropriate nematicide is not a permanent/long-lasting solution for nematode management. Many chemical nematicides are expensive; carcinogenic; and toxic to humans, animals and the environment. They are also known to contaminate groundwater and deteriorate soil quality eventually. Moreover, unfavourable climatic conditions can make the applied nematicide ineffective against nematodes [43]. Therefore, many of the chemical nematicides have already been banned or highly restricted worldwide [57]. A few chemical nematicides have been gaining a lot of importance in recent times due to their reduced toxicity on non-target soil organisms. Fluensulphone (heterocyclic fluoroalkenyl sulphone nematicide) is one such product that is reported to lack many of the drawbacks evident in other chemical controls [58]. Feist et al. [59] reported on the inhibitory effects of fluensulphone on *G. pallida* hatching and compared its efficacy with that of aldicarb, fluropyram and abamectin. Fluensulphone was also found effective against *M. javanica*, *M. incognita* and *M. arenaria* [58]. However, the long-term effects of this chemical have not been studied yet, and further research is essential to validate its non-toxic effects on the environment and human health.

Various conventional management practices include the biofumigation of affected crop fields, the use of clean plant material and agricultural equipment, heat treatment and crop rotation. The disinfection of seeds and bulbs in hot water or formaldehyde is also practised by many growers [60]. Multiple management strategies such as organic amendments, soil solarisation and chemical control have also been adopted to protect tomato plants against devastating *Meloidogyne* species [61]. Among these strategies, chemical fumigants such as Paladin (dimethyl disulfphide or DMDS), Telone II, Vapam (metam sodium) and chloropicrin were found successful to a certain extent but were very difficult to apply, had long re-entry periods, were very expensive and needed buffer zones [62]. Vegetable growers in Georgia, USA, found the application of chemical fumigants, combined with non-fumigant nematicides such as Vydate, Nimitz, Velum prime, Mocap, Movento, Salibro and Counter 20 G, to be very effective against *Meloidogyne* species [62]. Douda et al. [63] recently reported on the application of ethanedinitrile (EDN) as a promising fumigant controlling the spread and infection of *Meloidogyne hapla* and improving carrot yield in Litol, Czech Republic. The efficacy of the fumigant was found to be affected by various environmental conditions, such as temperature, humidity and soil properties. Moreover, due to the acute toxicity of EDN, untrained farmers were not allowed to fumigate the soils on their own [63]. It is expected that health and environmental issues would restrict the use of these chemicals in the long term.

Youssef and Eissa [64] have studied the role of biofertilisers in the management of PPN. These biofertilisers, in comparison to chemical fertilisers, have demonstrated a better effect on the control of RKN, as well as improving the plant growth and yield. Velasco-Azorsa et al. [65] recently characterised the nematicidal and phytotoxic activities of seven plant extracts against *Nacobbus aberrans*, the false root-knot nematode. The extracts from *Adenophyllum aurantium*, *Alloispermum integrifolium* and *Tournefortia densiflor* were found to have the best nematicidal potential [65]. Mokrini et al. [3] listed the application of various biofertilisers, based on plant extracts and botanical oils, which were found effective against PPN and against *Meloidogyne* species in particular. The application of biofertilisers to plants, soil or seeds can result in the nutrients becoming available to the plants and can therefore enhance plant growth and productivity [66]. Limitations on the use of biofertilisers can be environmental conditions like soil temperature, pH and moisture, as the microbes associated with biofertilisers are not be effective unless these factors are favourable [64]. Natural compounds, like neem extracts, have been widely used in the past due to their nematicidal properties against nematodes [67]. Neem is a natural herb, known to have many antibiotic characteristics due to its naturally occurring compounds, like azadirachtin, an antibacterial compound proven to control pests [68]. Secondly, the decomposition of neem leaves produces ammonia gas that acts as a fumigant to destroy microbes naturally [67]. Amendments obtained from the decomposition of natural products such as *Brassica* species [6], castor oil plant (*Ricinus communis*) and velvet bean (*Mucuna* spp.) have also proven to be very effective against nematode growth [69,70]. A study on *Brassica* species has shown 85–90% nematode mortality by incorporating *Brassica* leaf extracts into soil infested with potato cyst nematodes [6]. *Brassica* species contain glucosinolates that, on hydrolysis, produce biologically active compounds like thiocyanates and isothiocyanates that are capable of controlling pests to a significant extent. Moreover, these *Brassica* leaves on decomposition produce toxic volatile sulphur-containing compounds such as dimethyl disulphide and carbon sulphide that act as biofumigants to completely clean up soils [71]. However, most of the time, Brassicaceae plants serve as hosts for many devastating species of nematodes, such as *Meloidogyne* spp. [69] and hence, increase the nematode numbers instead of controlling them. Therefore, it is essential to study the host range of the target nematode(s) before selecting a plant for biofumigation. Although organic amendments are very successful at controlling nematode growth, a major limitation is that they are time-consuming to use with no immediate effects after application. The other major limitation of using organic amendments is their potential to stimulate a broad range of other organisms, for example nematode-trapping fungi or other potential predators or parasites of PPN [69,72,73]. In a relevant case study performed in China, an increase in omnivore and predator nematode levels were reported within 1 month of compost application to a corn crop but they disappeared after 2 months [74]. A study in Florida [70] illustrated the impact of applying organic amendments, such as green manure residues, sunn hemp amendment, yard waste compost, previous crop residues and composted municipal soil waste, and has reported the difficulty in maintaining one beneficial group of organisms in a particular soil. Sometimes, organic amendments on decomposition release volatile/non-volatile compounds that may hamper plant growth. In the case of ammonia-releasing organic amendments, soil microorganisms readily oxidise ammonia into nitrites and nitrates through the process of nitrification and further cause a decrease in soil pH, making the soil ineffective for plant growth [68]. In such cases, organic nitrification inhibitors may be needed to improve the efficiency of such organic amendments. Thoden et al. [75] highlighted the application of various organic amendments that have proven successful in mitigating the effects of PPN in various crop plants. The amendments, such as slurries and their organic acids, had the potential to accumulate/form high concentrations of nematicidal compounds and were able to create anaerobic conditions to directly suppress PPN population. Korthals et al. [76] reported the application of eight soil health treatments (anaerobic soil disinfestation, biofumigation, chitin, compost, grass-clover, marigold, a physical method and a combination of marigold, compost and chitin) to manage the root-lesion nematode *Pratylenchus penetrans*. All of them proved to be better alternatives to chemical treatments due to their beneficial influence on the physical and chemical properties of soil. The overall positive contribution of the organic amendments is due to their role in increasing the population of free-living nematodes, entomopathogenic nematodes and bacteria, which subsequently play an important role in stimulating plant growth, nutrient supply and mineralisation, thus making plants resistant to PPN infections [75].

In the past few years, according to the literature, it is evident that the use of biological-based products is safer and more sustainable compared to chemical treatments or methods [3,57,77,78]. In the last 15 years, there has been a thorough investigation of various microbial-based products. Dong [60] has reviewed a few commercial microbial products that were used in the early 21st century. Some of those products were abamectin (*Streptomyces avermitilis*-based), a *Paecilomyces lilacinusa*-based formulation (Biocon, Paecil and Miexianing), BioNem (*Bacillus firmus*-based), Deny (*Burkholderia cepacian*-based), DiTera (*Myrothecium*-based), Nemout (nematophagous-fungi-based), Royal 350 (*Arthrobotrys irregularisa*-based) and Xianchongbike (*Pochonia chlamydosporium*-based). All of them were proven successful in inhibiting, repelling, parasitising and killing PPN but had a common limitation of inconsistent field performance. Therefore, a thorough understanding of host–parasite interactions and long-term studies in the presence of those products would contribute to successfully and sustainably managing PPN infestations.

## 7. Fermentation and Microbial-Based Products

Fermentation-based products have gained a lot of interest due to their sustainable and cost-effective production methods [79]. For the large-scale production of fungi or any other biological materials, a combination of solid and liquid fermentation could be an effective approach [80]. Brand et al. [81]; Twamley et al. [82]; and Wu et al. [83] reported the effectiveness of fermentation extracts against PPN. Recent applications of fermentation extracts of *Myrothecium verrucaria* proved highly effective against *M. incognita* and *H. glycines* by inhibiting the egg-hatching process, and demonstrated its lethal effects on J2 juveniles of *B. xylophilus*, *M. incognita* and *H. glycines* [83]. Similar observations were recorded by Hallmann and Sikora [84] using filtrates containing strains of *Fusarium oxysporum*, which produced toxins that were effective against *M. incognita*. However, the organic compounds in the fermentation medium retarded plant growth.

Beneficial soil-borne organisms, so-called biological control agents (BCA), are known to act as antagonists against PPN [73]. Biological control agents of bacterial or fungal origin are known to control plant diseases and also improve plant productivity and immunity against various plant pathogens [85]. Zhang and Yang [80] have reported several commercial products, based on nematophagous fungi, that were capable of plant growth promotion and the reduction of damage caused by nematodes. Nematophagous fungi are those types of fungi that are capable of parasitising, capturing and paralysing nematodes at various growth stages of their life cycle. These nematophagous fungi have gained a lot of importance due to their specificity in targeting nematodes. They have various mechanisms to control nematodes; some of them use traps to seize the free-living nematodes; others use spores to parasitise nematodes, and some of them produce toxins to paralyse nematodes and eventually kill them, while some of them infect the eggs and cyst stages of nematodes using hyphal tips [80].

Bacterial and fungal endophytes such as *Pseudomonas aeruginosa*, *Pseudomonas fluorescens*, *Rhizobium etli*, *Pantoea agglomerans* and *Brevundimonas vesicularis* have also been reported to be suitable for the biological control of PPN [86]. These endophytes have the potential to colonise endorhizae at some stage in their life cycle. They not only help in plant growth promotion but are also reported to have antagonistic activity towards *M. incognita* and *G. pallida* [86]. Toxic metabolites produced by fungal filtrates after fermentation are known to show inhibitory activity towards sedentary nematodes, compared to their activity towards non-parasitic nematode species [84]. This could be due to the non-exposure of sedentary nematodes to high levels of toxins from the organic matter, whereas the non-parasitic nematodes are constantly exposed to them [84]. 

Table 4 lists some microbial and fermentation-based products that have been investigated and proven to manage PPN infestation in various plants.

## 8. Conclusions

The information in this review attempts to help provide an overall understanding of various aspects related to plant parasitic nematodes and their management. A thorough review of various management strategies of these pests demonstrates the limitations, importance and potential of biological materials in controlling PPN infection rates. To overcome the limitations, multiple long-term experiments must be performed within a greenhouse before field application, to ensure the efficacy and consistent performance of the products. The use of fermentation-based products would be beneficial to the environment and serve as a more sustainable approach towards pest management. The application of such fermentation-based products as nematicides could prove very promising and form an integral part of sustainable agriculture. The results from the two Irish studies are the first to report the occurrence of PPN in the Republic of Ireland. The most frequently occurring nematodes found in Teagasc, Oakpark, were the migratory parasite herbivores *Pratylenchus*; *Tylenchus*, the herbivorous epidermal/root hair feeders; and *Tylenchorhynchus*, the herbivorous ectoparasite. In Teagasc, Johnstown Castle, the ectoparasites *Merlinius* and epidermal/root hair feeders *Aglenchus* were the most abundant. *Helicotylenchus* spp., *Meloidogyne* spp. and *Pratylenchus* spp. were commonly found in both sites. Members of the genus *Pratylenchus* are root-lesion nematodes, mostly the endoparasites of Poaceae and Leguminosae, however with a wide host spectrum including Rosaceae and the herbaceous and woody plants (black currant, apple trees, strawberries, raspberries, etc.). They also exist as endoparasites in the roots of Cruciferae (Brassicas) family members. Root lesion nematodes can easily spread while moving soil through various means, such as soil adhered to equipment, plant products, vehicles and humans. Sometimes root lesion nematodes multiply in the roots of mustard or other plants belonging to Brassicas. This could be a reason for the increased relative abundance of *Pratylenchus* spp. in the second site where oilseed rape was grown. Therefore, it is very much essential to check the crops that are selected for biofumigation. It is evident from the results that a wide range of PPN are present in Irish soils; hence, there is a need to focus on future studies to prevent their spread and impact on crop yields. However, the analysis was performed only in two specific sites and needs to be repeated in other major agricultural areas.

## 9. Future Studies

Thorough investigations are necessary to analyse the potential of combining the various available biological and cultural practices, in an effort to reduce economic losses due to nematode damage.

## Figures and Tables

**Figure 1 plants-10-02352-f001:**
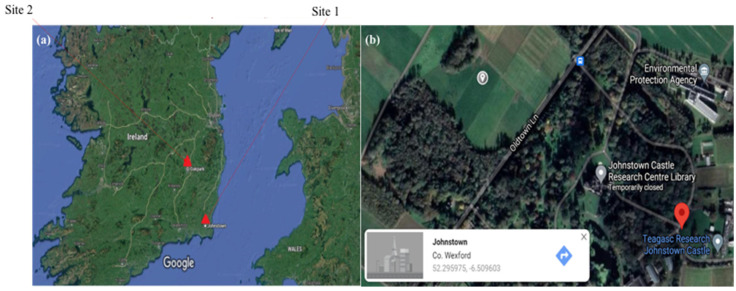
(**a**) Site locations in Republic of Ireland; (**b**) field sampling location of site 1, 52.295957 N & −6.509603 W, source Google maps.

**Figure 2 plants-10-02352-f002:**
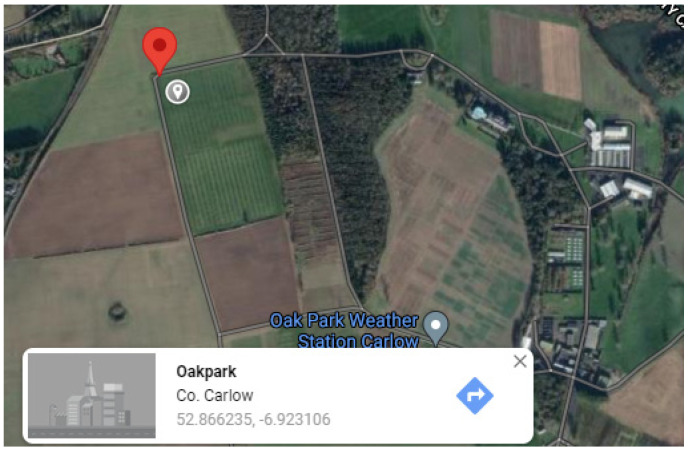
Field sampling location of site 2, 52.866235 N, −6.923106 W, source Google maps.

**Table 1 plants-10-02352-t001:** The mean relative abundance of PPN taxa found in the control group (n = 5) of site 1, including their standard deviations in parenthesis.

Nematode Taxa		Feeding Type	Mean Relative Abundance (±SD)
Family	Genus		
*Dolichodoridae*	*Merlinius*	Ectoparasites	6.90 ± (0.47)
*Tylenchidae*	*Aglenchus*	Epidermal/root hair feeders	6.05 ± (1.34)
*Tylenchidae*	*Neopsilenchus*	Epidermal/root hair feeders	2.07 ± (0.51)
*Longidoridae*	*Xiphinema*	Ectoparasites	1.39 ± (0.450)
*Hoplolaimidae*	*Helicotylenchus*	Semi-endoparasites	1.01 ± (0.31)
*Paratylenchidae*	*Pratylenchus*	Migratory endoparasites	0.98 ± (0.52)
*Longidoridae*	*Longidorus*	Ectoparasites	0.62 ± (0.24)
*Meloidogynidae*	*Meloidogyne*	Sedentary parasites	0.33 ± (0.03)
*Tylenchidae*	*Boleodorus*	Epidermal/root hair feeders	0.12 ± (0.01)
*Meloidogynidae*	*Meloidogyne*	Sedentary parasites	0.11 ± (0.01)
*Tylenchidae*	*Basiria*	Epidermal/root hair feeders	0.021 ± (0.00)
*Paratylenchidae*	*Pratylenchus*	Migratory endoparasites	0.015 ± (0.00)
*Tylenchidae*	*Tylenchus*	Epidermal/root hair feeders	0.008 ± (0.00)
*Tylenchidae*	*Lelenchus*	Epidermal/root hair feeders	0.002 ± (0.00)

**Table 2 plants-10-02352-t002:** The mean relative abundance of PPN found in the study plot (n = 4) of site 2, including their standard deviations in parenthesis.

Nematode Taxa		Feeding Type	Mean Relative Abundance (±SD)
Family	Genus		
*Telotylenchidae*	*Amplimerlinius*	Herbivores–ectoparasites	1.00 ± (0.00)
*Telotylenchidae*	*Bitylenchus*	Herbivores–ectoparasites	0.50 ± (0.70)
*Tylenchidae*	*Cephalenchus*	Herbivores–epidermal/root hair feeders	1.00 ± (1.00)
*Dolichodoridae*	*Dolichodorus*	Herbivores–ectoparasites	1.00 ± (0.00)
*Dolichodoridae*	*Dolichorhynchus*	Herbivores–ectoparasites	1.00 ± (0.00)
*Hoplolaimidae*	*Helicotylenchus*	Herbivores–semi-endoparasites	1.25 ± (0.22)
*Tylenchidae*	*Malenchus*	Herbivores–epidermal/root hair feeders	1.33 ± (0.29)
*Heteroderidae*	*Meloidogyne*	Herbivores–sedentary parasites	1.00 ± (0.00)
*Ironidae*	*Ironus*	Predators	1.00 ± (0.00)
*Pratylenchidae*	*Pratylenchoides*	Herbivores–semi-endoparasites	1.00 ± (0.00)
*Pratylenchidae*	*Pratylenchus*	Herbivores–migratory endoparasites	9.40 ± (1.32)
*Hoplolaimidae*	*Rotylenchus*	Herbivores–semi-endoparasites	2.00 ± (0.26)
*Trichodoridae*	*Trichodorus*	Herbivores–ectoparasites	1.00 ± (0.00)
*Belonolaimidae*	*Tylenchorhynchus*	Herbivores–ectoparasites	4.00 ± (0.00)
*Tylenchidae*	*Tylenchus*	Herbivores–epidermal/root hair feeders	7.33 ± (1.11)
*Longidoridae*	*Xiphinema*	Herbivores–ectoparasites	1.33 ± (0.29)

**Table 3 plants-10-02352-t003:** Classification of PPN groups according to genus, feeding type, physical manifestations and mode of action.

Nematode Groups	Genus	Feeding Type	Physical Manifestations	Mode of Action
Root-Knot	*Meloidogyne* spp.	Obligate	Forms galls (root-knots) on infected roots	Feeds on giant cells of the root and suppresses the host defence mechanisms
Cyst	*Heterodera* and *Globodera* spp.	Obligate biotrophs	Forms cysts (enclosing eggs) due to a large multinucleate feeding structure called the syncytium	Dissolves plant cell walls and fuses protoplasts Effectors target the host cell nucleus and suppress plant defence mechanisms.
Root lesion	*Pratylenchus* spp.	Polyphagous, migratory, intercellular root endoparasites	Formation of lesions, necrotic areas, browning and plant cell death, often followed by root rotting.	Secretions from pharyngeal glands have effectors that degrade plant cell walls.
Burrowing	*Radopholus similis*	Migratory endoparasite	Weakens the root system, forms dark lesions, root rotting and causes toppling disease	The effectors contain plant-cell-wall-degrading enzymes like cellulases, pectate lyases and xylanases.
Stem and bulb	*Ditylenchus dipsaci*	Migratory endoparasite	Causes stunted growth, twisted stems and the discoloration of bulbs	Feeds on the parenchymatous cells of the cortex and produces cell-wall-softening enzymes and effectors such as expansins
Pine wilt	*Bursaphelencus xylophilus*	Migratory endoparasite	Completely infects and kills all the pine trees growing in an area	Parasitizes plants with the help of cellulose-degrading proteins like glycoside hydrolase It is carried with the help of insect vector, *Monochamus* beetles.
Reniform	*Rotylenchulus reniformis*	Sedentary Semi-endoparasite	Leads to moisture and nutrient deficiency in infected host along with root necrosis, chlorosis and stunted growth	Feeds on pericycle and endodermal root cells by inserting 1/3 of their anterior body Cell walls break to form a two-cell-deep syncytium.
Large plant parasitic species	*Xiphinema index*	Ectoparasite	Infection retards root extension, causes swelling and gall formation	Have feeding mechanisms similar to root-knot nematodes
False-root knot	*Nacobbus aberrans*	Migratory juveniles Sedentary adult female	Causes cavities and lesions on root tissues; root galls are formed around feeding sites	Induces the partial dissolution the of cell wall and the fusion of protoplasts to form a syncytium
White tip disease variety	*Aphelenchoides besseyi*	Ecto/endo parasite Fungivorous	Infected plants have stunted growth, and other symptoms include chlorotic white tips on leaves, leaf necrosis, reduction in rice grain size and number	Does not induce re-differentiation of plant cells Local cell damage, tissue disintegration and browning in epidermal cells and palisade parenchyma

(Modified from Jones et al. [43]).

**Table 4 plants-10-02352-t004:** Bionematicides based on biological control agents and fermentation processes.

Fermentation Extracts/Microbial Based Products	Target Nematode	Plants	Reference
Fermentation filtrate of *Myrothecium verrucaria*	*M. incognita*, *H. glycines*, *B. xylophilus*, *Hirschmanniella* spp.	Soybean and tomato	[83]
*Paecilomyces lilacinus* produced through the solid-state fermentation	*Meloidogyne* spp.	Tomato	[79]
*Bacillus firmus*, commercial WP formulation (BioNem) (*WP:Wettable powder)	*M. incognita*	Tomato	[61]
Isolate of *Bacillus thuringiensis*	*R. reniformis* *M. incognita* *P. penetrans*	Tomatoes, pepper and strawberry	[87]
Abamectin (a fermentation product from *Streptomyces avermitilis*), a strain of *B. thuringiensis*, Nemaless (containing strains of *Serratia marcescens*)	*M. incognita*	Faba beans	[88]
Abamectin (a fermentation product from *Streptomyces avermitilis*)	*M. incognita*, *M. javanica*, *Radopholus similis*, *D. dipsaci*	Tomato, banana, garlic, cucumber	[89]
*Bacillus methylotrophicus* strain R2-2 and *Lysobacter antibioticus* strain 13-6	*M. incognita*	Tomato	[56]
Strains of *F. oxysporum*	*M. incognita*	Lettuce	[84]
*Trichoderma harzianum* and *Trichoderma lignorum* isolates	*M. javanica*	Tomato	[90]
DiTera (a fermentation product of the fungus *Myrothecium verrucaria*)	*P. penetrans*	Red raspberry (*Rubus idaeus*)	[91]
ACS5075 (proprietary blend of fermentation and plant extracts with micronutrients, Alltech, Inc., Nicholasville, KY USA) and ACS3048 (a microbial-based product)	*M. javanica, M. incognita*	Tomato	[92] [93]
Compost- Aid (*Lactobacillus plantarum, Bacillus subtilis* and *Enterococcus faecium*)	*Pratylenchus brachyurus* and *M. javanica*	Soyabean	[94]
*Bacillus megaterium* (ATCC No. 14581), *Pseudomonas fluorescens* (ATCC No. 13525), *Trichoderma viride* (MTCC No. 167), *Paecilomyces lilacinus* (PDBC PL55) and *Glomus intraradices*	*M. incognita*	*Withania somnifera* L.	[95]
BioNem (containing lyophilised bacteria spores of *Bacillus firmus*)	*Meloidogyne* spp.	Tomatoes	[78]
Micosat (by C.C.S. Aosta, Italy) (constituted by 40% roots hosting arbuscular mycorrhiza forming fungi of *Glomus* spp. and a mixture of antagonistic fungi, rhizobacteria and yeasts)	*M. incognita*	Tomatoes	[85]
Bioact (*P. lilacinus* and *Bacillus firmus*)	*Meloidogyne* spp.	Tomatoes	[96]
*Bacillus sphaericus* B43	*M. incognita*	Tomatoes	[86]

## Data Availability

Data has been stored in the library repository of Institute of technology Carlow, Ireland.

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
