# Peer review of "Plant Parasitic Nematodes: A Review on Their Behaviour, Host Interaction, Management Approaches and Their Occurrence in Two Sites in the Republic of Ireland"

_plants, 2021, doi:10.3390/plants10112352_

Round 1

Reviewer 1 Report

Unfortunately, I can not recommend this manuscript for publication.

  1. The title of this manuscript : Plant parasitic nematodes: a review on their behaviour, host interaction, management approaches and their occurrence in two sites in the Republic of Ireland” and contains 23 pages of text with tables and figures. However, only section: “Irish Case Studies” (5 pages) describes original results from the Republic, and all other sections deal with a review of literatures published by other authors and which are not based on results obtained in this region.
  2. Although authors extended the material and method section for original study, but it is not clear how authors identified nematodes using the SSU sequences.

Author Response

Manuscript has been revised as per the comments made by Reviewer 2 and 3.

Reviewer 2 Report

In the revised manuscript the authors have answered my queries and where appropriate amended the text. However, the reference list at the end needs careful checking, for example,

751: check Sitora authors, what are surnames for the initials SK and DA?

679: Manzanilla-Lopez  -- is this reference complete?

711: Nima et al --- is this complete?

768: is this complete?

800: only one reference to Wu but in text  -- 409: Wu et al 2020 and 413: Wu and Zhand 2020  --- are these the same reference?

Author Response

1.

751: check Sitora authors, what are surnames for the initials SK and DA?

The surnames were Schafer K. and Dababat A.A.

The reference has been corrected in the manuscript:

Sikora R.A., Schafer K. and Dababat A.A., 2007. Modes of action associated with microbially induced in planta suppression of plant-parasitic nematodes. Australas. Plant Pathol. 36, 124–134.

2.

679: Manzanilla-Lopez  -- is this reference complete?

The page number 223 was missing.

The reference has been edited in the manuscript

3.

711: Nima et al --- is this complete?

There was a mistake with this reference.

It has been corrected to in the manuscript:

Weerasinghe, R.R., 2004. Rhizobial-Legume Symbiosis and Root Knot Nematode Parasitism: Common Signal Transduction Pathways in Legumes. North carolina state university.

4.

768: is this complete?

The missing issue number (2) has been added in the manuscripts

5.

800: only one reference to Wu but in text  -- 409: Wu et al 2020 and 413: Wu and Zhand 2020  --- are these the same reference?

They are the same references.

The mistake has been corrected in the manuscript

Reviewer 3 Report

The authors were able to address all my previous comments, so I suggest to publish the manuscript in present form.

Author Response

This manuscript is a resubmission of an earlier submission. The following is a list of the peer review reports and author responses from that submission.

Round 1

Reviewer 1 Report

The title of this manuscript looks rather misleading. Readers expect to find a review of some research works on host-plant interaction and examples of control nematode measures conducted in Ireland and the list of nematodes found in Ireland, but instead, it deals with very general review of some studies on nematodes, which are rather well-known from nematology text books and many comprehensive review articles. Conclusions are very general and not original and they should not contain any references. I do not recommend this manuscript for publication in its present form.

Other comments.

1.There is no any sense to list the major plant parasitic nematodes in such manuscript. The list of plant parasitic nematodes found in Ireland should be given. For example,  Anguina pacificae (see Fleming et al., 2015. Journal of Nematology 47, 97-104), M. naasi, M. minor, M. fallax and others.

  1. Original Information on findings of plant nematodes from two sites should be given in other format with indication of specimen numbers per soil volume or weight. Methods of extraction and nematode identification and results should be described in details.

Reviewer 2 Report

In the ms the most interesting contributions are the sections ‘Current Management Strategies and Their Limitations’ and ‘Fermentation and microbial-based products’  with the very informative Table 5. (pages 13-19).

The section: ‘Plant Parasitic nematode species and their distribution’ and the data in Table 1 are  mostly the repetitions of the data in paper published by Jones et al. 2013 that is cited in the ms.

The section ‘Plant Responses to Nematode Infection’ with Table 4 (as mentioned in the ms: modified from Jones et al., 2013) contain mostly the data already reviewed by Jones et al., 2013.

= The part ‘Irish Case studies’ with a lists of fauna in two sampling sites does not correspond to the REVIEW format of the paper. It is the Original Research Paper inside the Review. This paper has to be excluded from the Review and prepare separately for the publication; now it is not good described and not good sectioned. Tables 2 and 3 in this section are to be revised because: i) they are based on different methods: Table 2: on the molecular analyses vs Table 3 on morphological analyses. ii) they are designed with different columns: in Table 3 there are names  of nematode families and genera vs in Table 2 only names of species without their attributions to the families. In addition Table 2 and Table 3 contain several errors. It is necessary to mention what taxonomic classification of the Nematoda phylum was used by authors; and what the classification of feeding types the authors used for different nematode genera. Usually for the feeding types characterization the researchers use the following paper:

Yeates G.W. et al., 1993. Feeding Habits in Soil Nematode Families and Genera-An Outline for Soil Ecologists. Journal of Nematology 25(3):315-331.

But in tables 2 and 3 there are some deviations from the generally accepted classification of the feeding types.

Table 2 and 3: Aphelenchoides = migratory endoparasites

= (Yeates et al.,  1993): Aphelenchoides (leaf and bud nematodes)… feed on fungi and on aerial parts of plants

Table 3: Ironidae: Ironus - Epidermal cell feeders

= (Yeates et al., 1993): Ironidae: Ironus – predacious nematodes + unicellular eucaryote feeders

Taxonomic classification, as it used in Table 3:

Table 3: -Bitylenchus (fam. Tylenchidae)

= However, the Bitylenchus belongs to the fam. Telotylenchidae (see Van Megen et al. 2009. A phylogenetic tree of nematodes based on about 1200 full-length small subunit ribosomal DNA sequences. Nematology, 2009, Vol. 11(6), 927-950

Table 3: Amplimerlinius (fam. Dolichodoridae)

= However, Amplimerlinius belongs to the fam. Telotylenchidae (see Van Megen et al. 2009)

Table 3:  Pratylenchoides (fam. Paratylenchidae)

= However, Pratylenchoides  belongs to the fam.Pratylenchidae.

Other notes:

L 100

Figure 1: a) Site locations in Republic of Ireland b) Field sampling location of site 1, 52.295957N & -100 6.509603 W, source Google maps.

= However, in Figure 1 a  the  GIS data are different: 52.295975N

L 153

These egg masses occur as clusters in the form of galls on the root surface

= Galls are not the egg masses, but the expanded root parts containing root-knot nematode females, or pathological nodes on roots induced by traumatic wounds done by the nematode stylets or spears puncturing the root surface

L 228 and page 11: text and Table 4.

Ditylenchus dipsaci is characterized as migratory endoparasite (Table 4) whereas ‘the rice-stem nematode, Ditylenchus angustus feed ecoparasitically on the leaves and stems of rice (Line 239)’

= What the HPR mode of the Ditylenchus spp.: it is different among species, ecto- and endoparasites??

L 382

ectoparasitic nematode Pratyenchus.

=Pratylenchus (ENDOparasitic) or Paratylenchus (ECTOparasitic)???

L 386

Members of the genus Pratylenchus are root-lesion nematodes, mostly parasitic to wheat,

=Pratylenchus spp. are mostly the endoparasites of  Poaceae and Leguminosae, however with wide host spectrum including Rosaceae and the herbaceous and woody plants (black currant, apple trees, strawberries, raspberries, etc.), and these nematodes are endoparaistes of roots of the Cruciferae (Brassicas), as you mentioned in the ms, Lines 392-394.  

Reviewer 3 Report

This is a comprehensive review on the occurrence, host interactions and management approaches to plant parasitic nematodes (PPNs) found in specific sites in Ireland. The work is clearly presented. I have only a few comments/queries.

97: explain significance of using 18SrRNA gene

105: explain what you mean by Molecular Operational Taxonomic Units

Can the authors explain how the values in the third column of Tables 2 and 3 relate.

121: in legend for Table 2, explain what is meant by Mean Relative Abundance (in addition to text 108)

Table 5 legend: put Biological Control Agents in full

Table 5: Micosal, any further information on what it contains?

254: perhaps mention recent work on fluensulfone as a potential chemical PPN nematicide - Pest Biochem. Physiol. 165:104541 - and there may be other examples.

337: Yan et al (2020) is not in reference list. I have not checked all the refs and so please check. 

399-407: suggest for clarity the authors head this paragraph -Future Studies - and use bullet points.

505: Galadima et al: line 506: is the reference complete, for example after 2, should there be 1101779?

Minor typo: 104/5: Data are pleural - were expressed

Reviewer 4 Report

I would adjust title as "occurrence on two sites in Ireland", from the present tittle it seems than occurrence in whole country was evaluated.

Table 1:

Ditylenchus africanus does attack peanuts, not potatoes, it is not a problem in Europe

B. xylophilus is present also in Europe (Portugal, Spain)

I would add Ditylenchus dipsaci into this table.

Lines 142 - 143: "These nematodes feed with the help of specialized cells present around the female head" - this is true for sedentary endoparasites only, not for ectoparasites or migratory endoparasites.

Line 186: I am not sure if 20 years period that cysts can survive in soil is true for Heterodera sp., eg Heterodera schachtii does not survive in soil so long, please check this.

Line 260: Chemical control of Meloidogyne - it would be good to mention recent research on ethandinitrile (eg Douda, O., Manasova, M., Zouhar, M., Hnatek, J., & Stejskal, V. (2021). Field validation of the effect of soil fumigation of ethanedinitrile (EDN) on the mortality of Meloidogyne hapla and carrot yield parameters. Agronomy, 11(2), 208.)

Line 379 - 380: "...first to report the occurrence of PPN in the Republic of Ireland." Did you mean first results of research of complete nematofauna?